# Isolated Spleen Metastases of Endometrial Cancer: A Case Report

**DOI:** 10.3390/medicina58050592

**Published:** 2022-04-26

**Authors:** Marko M. Stojanovic, Vesna Brzački, Jelena D. Zivadinovic, Nebojsa S. Ignjatovic, Marko D. Gmijovic, Miodrag N. Djordjevic, Ilija Golubovic, Nada G. Nikolić, Novica Z. Bojanic, Miroslav P. Stojanovic

**Affiliations:** 1Gastroenterology and Hepatology Clinic, University Clinical Center, University of Niš, 18000 Niš, Serbia; marcss994@gmail.com (M.M.S.); brzackiv@gmail.com (V.B.); 2Clinical Center Niš, Bul. Dr Zorana Djindjica 48, 18000 Niš, Serbia; 3Clinic for Anesthesiology, University Clinical Center, University of Niš, 18000 Niš, Serbia; jelena5491@gmail.com; 4Digestive Surgery Clinic, University Clinical Center, University of Niš, 18000 Niš, Serbia; n.ignjat@gmail.com (N.S.I.); markogmija@gmail.com (M.D.G.); mija.djordjevic@yahoo.com (M.N.D.); golubovicilija@yahoo.com (I.G.); 5Clinic for Anesthesiology, University Clinic RWTH Aachen, 52074 Aachen, Germany; nnikolic@ukaachen.de; 6Pathological Physiology Institute, Medical Faculty, University of Niš, 18000 Niš, Serbia; bojanicnovica@gmail.com

**Keywords:** spleen, metastases, endometrial, cancer, splenectomy

## Abstract

Background: Isolated splenic metastases from endometrial cancer, which is a relatively common malignancy, are extremely rare findings; to date, only 14 cases have been reported in the literature. Case Summary: We report a patient with isolated splenic metastases of endometrial cancer 3 years after radical surgery of the primary tumor. The patient was successfully treated by splenectomy and six cycles of paclitaxel. Fifty months after splenectomy, she was alive and well, and with no evidence of disease. Conclusion: Isolated spleen metastasis of endometrial cancer is very rare. Radical surgery and adjuvant therapy may offer excellent long-term survival.

## 1. Introduction

The spleen is not a common site for metastasis of solid, non-hematological tumors, primarily due to its specific anatomy and microenvironment. Most metastases to the spleen generally occur due to multivisceral metastatic cancer dissemination but are rare as a solitary lesion [1] from ovarian [2,3], colorectal [4], and lung malignancies [5]. Isolated splenic metastases from endometrial cancer, which is itself a relatively common malignancy, are extremely rare findings. Only 14 cases have been reported in the literature [6,7,8,9,10,11,12]. These metastases are usually discovered after a latent period of 11 to 120 months after the surgery for primary malignancy of the uterus. In most cases, they are the first sign of the primary tumor recurrence [10]. However, there are opinions that isolated splenic metastases indicate a better prognosis [11].

Herein, we report a patient with isolated splenic metastasis of endometrial cancer 3 years after radical surgery of the primary tumor. A literature search of online databases and search engines was performed (PubMed/Medline, Web of Science, and Embase) completed by an internet-based search. The keywords used were “endometrial, cancer, spleen, metastases”. All references of included articles were manually screened to find relevant articles.

## 2. Case Presentation

### 2.1. Chief Complaints

A 62-year-old woman was admitted to our hospital for vague abdominal symptoms, dyspepsia, and mild pain in the left hypochondrium.

### 2.2. History of Past Illness

The patient had a history of total hysterectomy with bilateral salpingo-oophorectomy 32 months prior due to stage pT1apN0 papillary serous form of endometrial adenocarcinoma, without subsequent adjuvant therapy. The patient was treated in another institution and referred to our institution—tertiary abdominal surgery center. We had only written data about her treatment—without adjuvant therapy after the first operation.

### 2.3. Physical Examination

Clinical examination was normal, except slight tenderness in the left hypochondrium with no clinical visceromegaly registered.

### 2.4. Laboratory Testing

The laboratory data (complete blood count; electrolyte status; and renal, hepatic, and pancreatic function tests) were within normal ranges. Tumor markers levels were also normal (cancer antigen 125: 22.6 U/mL; carcinoembryonic antigen: 2.3 U/mL; carbohydrate antigen 19-9: 16.8 U/mL).

### 2.5. Imaging Examination

Ultrasonography of the abdomen revealed an enlarged spleen (140 mm in total diameter) with two hypoechogenic, cystic-like, well-demarcated lesions located on the lower (76 mm × 62 mm) and upper (37 mm × 37 mm) poles of the spleen. T1-weighted magnetic resonance imaging also showed two well-circumscribed round hypointense lesions, similar to dimensions registered on ultrasonography located on the upper and lower poles of the spleen. Lesions were bright and hyperintense on T2-weighted magnetic resonance imaging, with peripheral signal enhancing and small endophytic lobulization of the bigger lesion, mimicking some form of the membranous component.

## 3. Multidisciplinary Expert Consultation and Final Diagnosis

The suspicion of spleen hydatid disease was raised by the radiologist. Immunohistochemistry and immunofluorescence serology were performed, and they were both negative. We indicated surgery under the tentative diagnosis of a cystic neoplasm of the spleen, possibly metastatic from previously treated endometrial cancer.

## 4. Treatment

Left subcostal laparotomy revealed an enlarged spleen surrounded with dense adhesions (Figure 1).

Total abdominal exploration was carried out, but it yielded no other pathological findings. There was no regional lymphadenopathy. The patient underwent a classic splenectomy. Spleen dimensions were 14 cm × 12 cm × 9 cm, and it weighed 650 g. The spleen capsule was intact, but a large part of the parenchyma was involved by two solid and partly colliquated necrotic tumors with irregular borders. The dimensions were similar to the preoperative measurements. Surgical biopsies were fixed in 10% formaldehyde overnight, processed in paraffin wax, and cut at 4 μm. The sections were stained with hematoxylin and eosin, periodic acid—Schiff (PAS)—Alcian Blue, van Gieson, and immunohistochemical avidin biotin complexes. Microscopic examination of the splenic tumor confirmed metastatic adenocarcinoma compatible with endometrial origin (Figure 2).

The immunohistochemical examination was positive for p53 (Figure 3) and vimentin WT-1 in the tumoral cells (Figure 4) but negative for estrogen receptor (Figure 5). These findings were similar to the immunohistochemical examination of the hysterectomy piece from the previous surgery. The postoperative recovery was uneventful, and the patient was discharged on the 11th postoperative day.

Subsequently, the patient was administered paclitaxel (175 mg/m^2^) every 21 d.

## 5. Outcome and Follow-Up

Fifty months after splenectomy and six cycles of paclitaxel, the patient was alive and well and without evidence of disease.

## 6. Discussion

Most metastases in the spleen generally occur due to multivisceral metastatic cancer dissemination and rarely as a solitary lesion [1]. Articles about isolated splenic metastasis are very rare, with less than 100 such cases reported in the literature [9,12]. There are several hypotheses that might explain the lower frequency of metastases in the spleen. Constant blood flow, rhythmic contraction of the spleen, the sharp angle of splenic artery branching, the lack of a lymph system, and the microenvironment of the spleen (e.g., the abundance of phagocytes in the spleen) are the factors emphasized in the majority of hypotheses [1,8,9,10,11,12]. However, influence of both anatomic factors and the suppressor effect of the splenic microenvironment to the resistance to metastases is still considered.

Reported cases are mostly from ovarian [1,2], colorectal [4], and lung malignancies [5]. Half of these are from the female genital tract, most commonly from ovarian malignancies (more than 30 cases) and the remainder from endometrial (14 cases), cervical (6 cases), and tubal (1 case) carcinomas [9,10,11,12]. Endometrial carcinoma is one of the most common malignancies and the most common genital tract cancer among women in developed countries [13]. Despite the possibility of early diagnosis and successful treatment options, 10–30% of patients develop recurrent disease [11,13]. Isolated splenic metastases from endometrial cancer are extremely rare [6,7,8,9,10,11,12]. They are usually solitary, limited to the splenic parenchyma, and occur as a result of hematogenous metastases in contrast to predominantly transperitoneal spread from ovarian cancer, thus rarely yielding solitary splenic metastases [2].

Splenic metastases occur as a synchronous or metachronous to the primary tumor. In the case of isolated splenic metastases of endometrial cancer, almost all published articles reported metachronous metastases discovered after a latent period (11 months to 120 months after the surgery for the primary malignancy of the uterus, with mean interval of 40.7 months and median of 28 months) [12]. Pissarra et al. recently reported a first case of synchronous isolated spleen metastases of the endometrial cancer [14]. In our patient, the latent period was 32 months. More than 60% of splenic metastases are asymptomatic when isolated, as in our patient. However, splenic metastasis can present as fatigue, weight loss, fever, abdominal pain, splenomegaly, or hematological disorders such as anemia or thrombocytopenia [1,8,12]. They are often incidentally discovered by ultrasonography, computed tomography, or magnetic resonance imaging procedures during routine postoperative follow-up of the patients.

Furthermore, other than conventional radiological techniques, 18-fluorodeoxyglucose positron emission tomography scanning can be used for confirmation of diagnosis [11]. Serum levels of tumor markers, such as carcinoembryonic antigen and carbohydrate antigen 19-9, might be preferable to predict the appearance of isolated splenic metastases [12]. Recently, fine-needle aspiration percutaneous biopsy has become a useful and frequently used method for the diagnosis of isolated splenic metastasis [13,15], but we did not perform this because of the accompanied risk of bleeding and possible spillage of cystic lesion content.

Treatment of isolated splenic metastasis includes several procedures. A splenectomy is indicated if there are isolated metastases or to reduce the tumor burden before chemotherapy [16]. Piura et al. [9] reported splenectomy to be an appropriate treatment in their patient with splenic metastasis from endometrial carcinoma. Splenectomy is also the treatment option for solitary splenic lesions from different primary tumors. Tang et al. [5] reported that splenectomy showed good effects in the case of isolated splenic metastasis from lung cancer. Laparoscopic splenectomy is a safe and feasible modality for the treatment of benign and malignant disorders of the spleen [17,18]. The results of the studies of laparoscopic splenectomy in patients with isolated splenic metastasis were successful [19,20], including the least invasive single-port laparoscopic or robotic approach [21,22]. However, intraoperative handling of the spleen might be challenging because of its often-considerable dimensions due to the presence of the tumor. The manipulation has to be cautious to avoid tumor rupture [12]. Therefore, we performed open splenectomy instead of laparoscopic, in which the spleen was fragmentized and stored in the endo-bag to be extracted [1]. Treatment options after splenectomy include chemotherapy (platinum based, paclitaxel, doxorubicin- single, doublet, or triplet), oral progestin, irradiation to the splenic bed, and both irradiation to the splenic bed and oral progestin [9]. Our patient was treated with six cycles of paclitaxel. After 50 months, she was alive and well, and with no evidence of disease.

## 7. Conclusions

Isolated spleen metastases of endometrial cancer represent a very rare condition. Radical surgery and adjuvant therapy may offer excellent long-term survival. Further studies analyzing the patients with isolated splenic metastasis are necessary for better understanding of the pathogenesis, clinical course, and treatment of this rare condition.

## Figures and Tables

**Figure 1 medicina-58-00592-f001:**
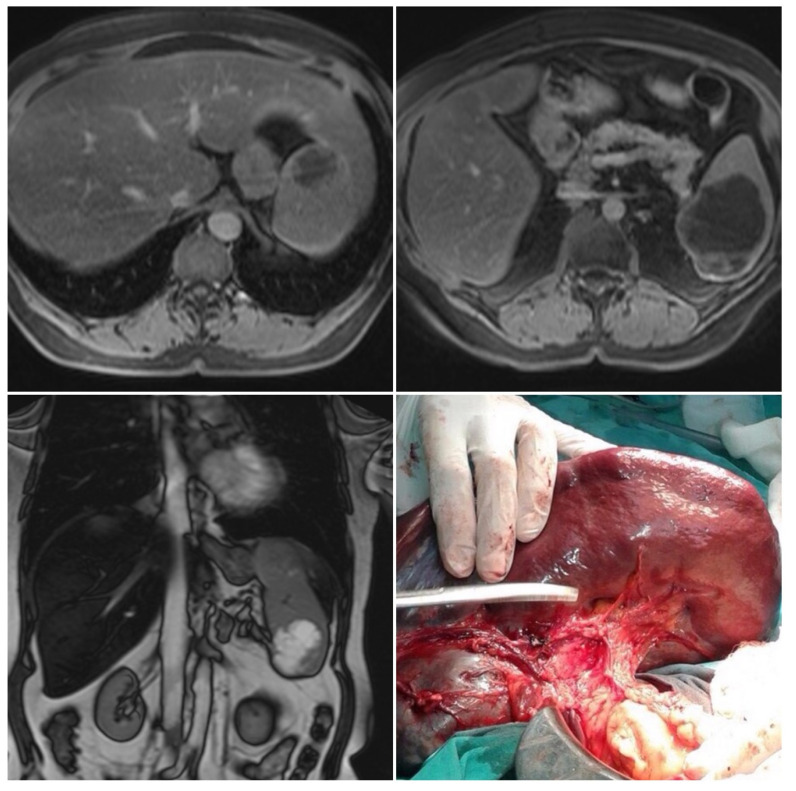
*Magnetic resonance imaging and intraoperative finding.* Two well-circumscribed round lesions of the spleen, T1 weighted—hypointense, T2 weighted—bright, and hyperintense—with peripheral signal enhancing and small endophytic lobulization of the bigger lesion.

**Figure 2 medicina-58-00592-f002:**
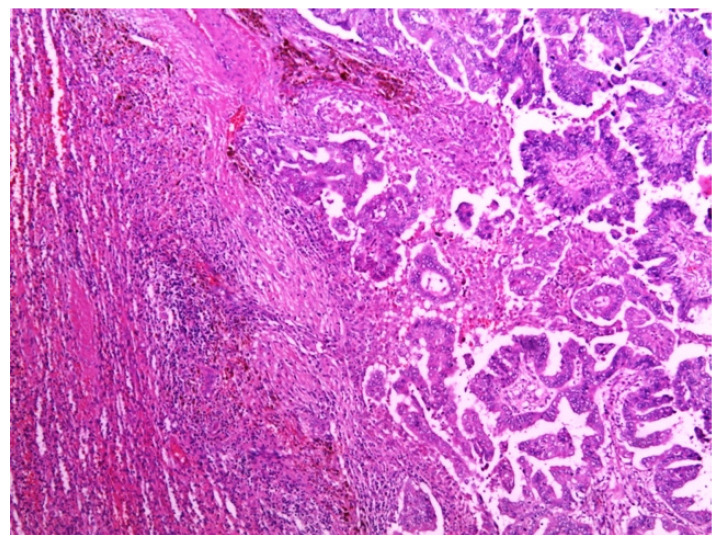
Metastatic deposit of tumor in the spleen, composed of papillae that were lined by cytologically high-grade neoplastic cells. Hematoxylin and eosin stain. Magnification: 50×.

**Figure 3 medicina-58-00592-f003:**
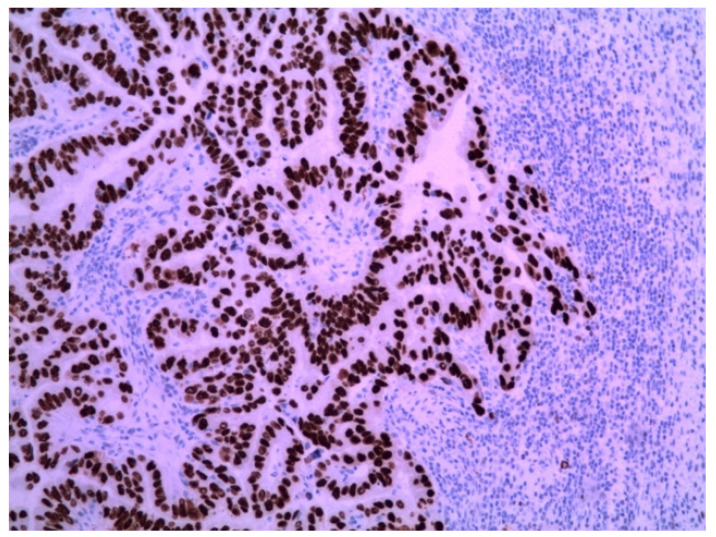
Metastatic endometrial papillary serous carcinoma. Strong diffuse p53 nuclear immunoreactivity was observed in the neoplastic cells. Magnification: 100×.

**Figure 4 medicina-58-00592-f004:**
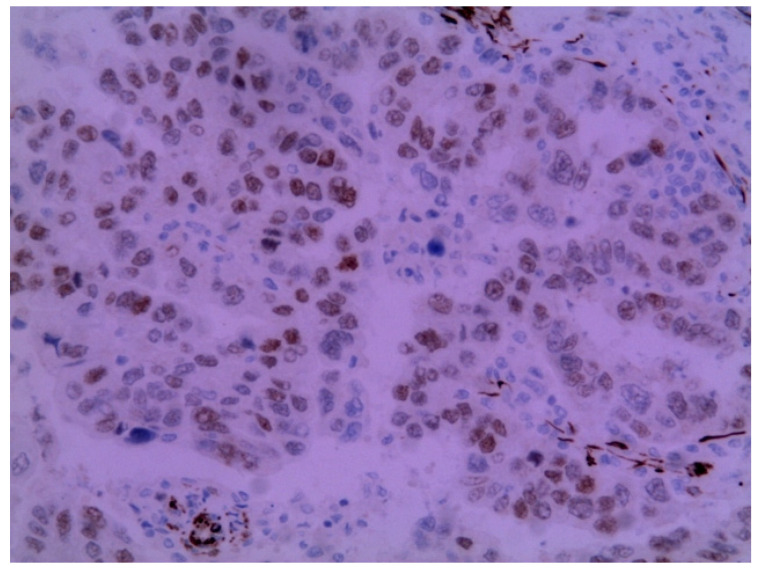
Metastatic endometrial papillary serous carcinoma. Diffuse moderate to focal strong nuclear immunoreactivity in the neoplastic cells to the WT1 protein was observed. Magnification: 200×.

**Figure 5 medicina-58-00592-f005:**
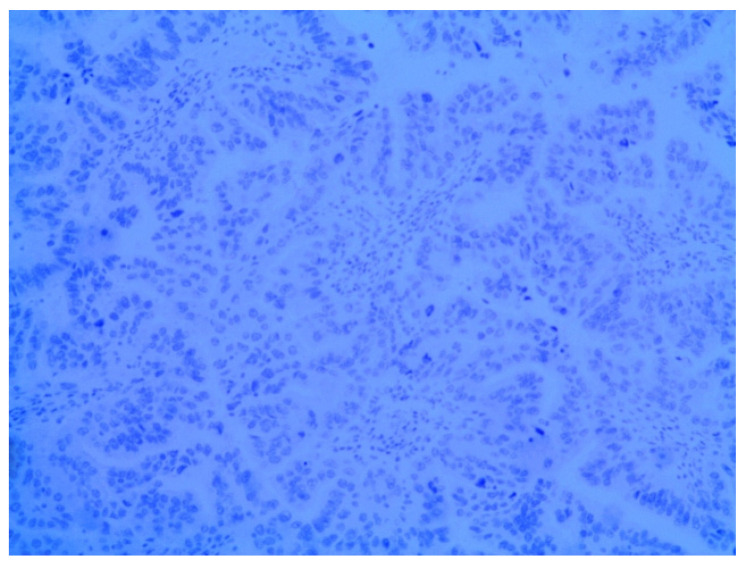
Negative nuclear immunoreactivity in the neoplastic cells to estrogen receptor α. Magnification: 100×.

## Data Availability

Not applicable.

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
