# Peer review of "Isolated Spleen Metastases of Endometrial Cancer: A Case Report"

_medicina, 2022, doi:10.3390/medicina58050592_

Round 1

Reviewer 1 Report

Marko M Stojanovic and coworkers report in this article “Isolated spleen metastases of endometrial cancer: A case report”. In this study, the authors reported and treated the spleen metastases after  3 years of radical surgery of endometrial cancer. The isolated spleen metastases were successfully treated by splenectomy and six cycles of paclitaxel. After 50 months, The patient was well and alive with no evidence of disease.

The overall study was presented well, and adequate experiments were conducted to assess the effectiveness of treatment on spleen metastases. The results were exciting and this study will be very useful for the endometrial cancer research community as well as “Medicina” readers.

Minor comments,

  • The abstract must be explained in detail. A detailed introduction should be explained.
  • Materials and methods section to be added in the MS and include the experimental details for Hematoxylin and eosin staining and other studies as well.
  • I recommend adding laboratory test results to the MS.
  • In the imaging examinations section was mentioned that ultrasonography of the abdomen, No images were added in the MS. I suggest adding them to the MS.

Author Response

Dear Reviewer 1, 

Thank You very much for Your suggestions . We will try to answer Your remarks: 

  1. Section Material and methods are not common in the case reports. We described, of course methods of pathological exam: "  Surgical biopsies were fixed in 10% formaldehyde overnight, processed in paraffin wax, and cut at 4 μm. The sections were stained with hematoxylin and eosin, Alcian Blue-Periodic acid-Schiff, van Gieson, and immunohistochemical avidin biotin complexes".
  2. We noticed that all biochemistry and hematology laboratory test results (over 50), including tumor marker levels  were normal. I think it will be too large and unnecessary  for the brief report of a case. 
  3.  Unfortunately, we have no  ultrasonography of the abdomen images. 

Sincerely Yours

Reviewer 2 Report

Authors wrote that the patienthad a history of total hysterectomy with bilateral salpingo-oophorectomy . Staging omentectomy should be considered in serous carcinoma. Furthermore, In non-EEC (apparent stage I), lymphadenectomy is recommended. Was the patient properly staged? The info is missing.

Fig. 2,3,4 are considerably big. Is it possible to create just one figure? 
minor:

- Please follow the reference list guidelines

Author Response

Dear Reviewer 2,

Thank You for the suggestions. We will try to answer on Your remarks: 

  1. Primary operation was performed in other institution. We have data only that staging omentectomy  was performed, and the patient staged -Ia.
  2. Figure 2,3,4 will be compacted by technical editor. 

Sincerely Yours

Reviewer 3 Report

In their manuscript “A Isolated spleen metastases of endometrial cancer: A case report” Stojanovic et al. intended to provide an understanding of the rare findings of Isolated spleen metastasis of endometrial cancer by registering a case report.

Following the case report analysis, the authors described what they found:

  1. The history, Physical examination, laboratory testing, imaging examination, and interesting diagnosis outcome.
  2. They have performed Histology analysis; immunohistochemical analysis of p53, WT1, and estrogen receptors.

This constitutes a comprehensive, and interesting rational body of work that is appreciable. However, there are a number of minor concerns that can be resolved to improve the quality of the manuscript, listed below.

  1. Please include the methods and materials used in this study.
  2. Please provide a brief summary of 14 previous cases of spleen metastases of endometrial cancer.
  3. A repetitive sentence in 3 sections “to date, only 14 cases have been reported in the literature”. Line 28, 44, and 159. Please rephrase the sentence.
  4. Rewriting the discussion portion is highly advisable with the consideration of the Therapeutic strategy in particular.

Author Response

Dear Reviewer 3, 

Thank You very much for Your suggestions . We will try to answer Your remarks: 

  • Separate section Material and methods are not common in the case reports. We described, of course, methods of pathological exam: "  Surgical biopsies were fixed in 10% formaldehyde overnight, processed in paraffin wax, and cut at 4 μm. The sections were stained with hematoxylin and eosin, Alcian Blue-Periodic acid-Schiff, van Gieson, and immunohistochemical avidin biotin complexes".
  1. Brief summary of 14 previous cases of spleen metastases of endometrial cancer will be added.
  2. A repetitive sentence in 3 sections “to date, only 14 cases have been reported in the literature”. Line 28, 44, and 159 will be rephrased
  3. We are mostly surgeons and we wrote in details on the most radical -surgical strategy options. Of course,  we will rewrite and enlarge other (adjuvant) therapeutic modality.

Sincerely Yours

Reviewer 4 Report

            The MS by Dr. Stojanovic et al. could be an appreciated account on a certain gynecologic oncologic rarity if it were not carelessly prepared and edited. My only current recommendation can be to reconsider it after a major revision, preferably by the hands of a native English speaker.  

            Major issues:

- why did a patient with serous endometrial carcinoma not receive any adjuvant therapy after her primary surgery ? this was a Type II tumor;

- what were the objective means of reviewing the literature ? Because, in the MS, the most recent citation given is from 2015, and I easily found a more recent one (from 2019) …

            Minor:

Lines 22-23, what does ‘Corresponding Author’s Membership in Professional Societies’ have in common with this MS ? 

Lines 30-32, there is a discordance between descriptions ‘3 years after radical surgery’ and ‘After 50 months, … ’; 

Line 36, the keyword ‘Isolated’ is not in line with MeSH recommendations;  

Line 48, please add the missing article before ‘better’;

Line 68, the word ‘hydroelectrolyte’ is not English;

Line 77, correct the lettering in ‘weigthed’;  

Line 86, use plural for the verb;  

Line 95, the legend to Figure 1 is too short; it should draw attention to particular findings on presented scans and the intraoperative picture‘; 

Line 98, please add the missing article before ‘classic’;  

Line 138, here the reader learns that the patient had ‘the lung nodules’; (?)

Lines 153 and 159: this same information is repeated;  

Lines 197-199; details of the performed spleen procedure are given at the end of the Discussion, and not in Lines 98-99;  

Line 204: ‘Conclusion’ includes 3 sentences;  

Refrences - there is a lack of order and consistency in how the references were reported;  

Why was the reference: ‘Pissarra AP, Cunha TM, Mata S, Félix A. Synchronous splenic metastasis of endometrial carcinoma. BMJ Case Rep 2019; 12: e230957.’ not traced and cited ?  

Author Response

Dear Reviewer 4,

Thank You very much for Your suggestions . We will try to answer Your remarks: 

  • Why did a patient with serous endometrial carcinoma not receive any adjuvant therapy after her primary surgery ? this was a Type II tumor;
    • The patient was treated in other institution and referred in our (institution -tertiary abdominal surgery center). We had only writing data about her treatment --without adjuvant therapy after first operation. 
  • - what were the objective means of reviewing the literature ? Because, in the MS, the most recent citation given is from 2015, and I easily found a more recent one (from 2019) …
    • Literature search using terms:  endometrial, cancer, spleen, metastases...... showed only 14 cases of isolated spleen metastases and one paper is not cited by technical error : Pissarra AP, Cunha TM, Mata S, Félix A. Synchronous splenic metastasis of endometrial carcinoma. BMJ Case Rep 2019; 12: e230957.’ This is very important, the only one case of the 14 cases of  synchronous isolated spleen metastases of the endometrial cancer. It will be cited under these circumstances. 

            Minor:

All errors will be corrected. 

Lines 22-23, what does ‘Corresponding Author’s Membership in Professional Societies’ have in common with this MS ? -deleted

Lines 30-32, there is a discordance between descriptions ‘3 years after radical surgery’ and ‘After 50 months, … ’; -corrected

Line 36, the keyword ‘Isolated’ is not in line with MeSH recommendations;  -but, this keyword is crucial and most descriptive in our opinion

Line 48, please add the missing article before ‘better’; -OK

Line 68, the word ‘hydroelectrolyte’ is not English; -OK

Line 77, correct the lettering in ‘weigthed’;  -OK

Line 86, use plural for the verb;  -OK

Line 95, the legend to Figure 1 is too short; it should draw attention to particular findings on presented scans and the intraoperative picture‘; -OK

Line 98, please add the missing article before ‘classic’;  -OK

Line 138, here the reader learns that the patient had ‘the lung nodules’; (?)-OK

Lines 153 and 159: this same information is repeated; -OK 

Lines 197-199; details of the performed spleen procedure are given at the end of the Discussion, and not in Lines 98-99;  -OK

Line 204: ‘Conclusion’ includes 3 sentences;  OK 

Sincerely

Round 2

Reviewer 4 Report

            I have read with attention both the MS by Dr. Stojanovic et al. in its corrected version and the rebuttal offered by the Authors in relation to the raised issues. Unfortunately, they did NOT respond to both major issues indicated by this reviewer, namely:

- why did a patient with serous endometrial carcinoma not receive any adjuvant therapy after her primary surgery while having had a Type II tumor?;  

- how was the literature searched ? (I expected here features of a systematic review methodology so that someone could repeat their search.)  

Yes, the Authors corrected the majority of indicated minor issues, yet did not correct many other points not indicated before. The MS was not checked by a native English speaker.  

All in all, this work is still far from a commendable level and I am in the only position to recommend: decline. 

            Just for example, SOME minor issues still not resolved:

- Line 70, the keyword ‘Isolated’ is not in line with MeSH recommendations: : https://meshb.nlm.nih.gov/search;  

- Line 79, is: ‘ … in the literature[6,7,8-12].’; why not ‘ … in the literature[6-12].’; 

- from Line 87 on, why is the ‘Case Presentation’ divided into subchapters and not one paragraph ?

- Line 102, the phrase ‘hydro electrolyte status’ is not English;

- Line 141, is: ‘ … Alcian Blue-Periodic acid-Schiff, … ‘; the literature rather refers to this staining as: periodic acidSchiff (PAS)Alcian Blue;  

- Lines 169-170: the number of CTX cycles is not given;

- Line 205, ‘sinchronous’ is misspelled.  

Author Response

  • Dear editor of the Medicina
  • Reviewer 4,

We are tried to improve our work based on the suggestions of 4 reviewers. After the 1 st round, fourth reviewer still has negative recommendation: “All in all, this work is still far from a commendable level and I am in the only position to recommend: decline.”

We are disappointing, and urged to You -please read careful his / her last remarks and our answers:  

  1. Why did a patient with serous endometrial carcinoma not receive any adjuvant therapy after her primary surgery while having had a Type II tumor?;  

  • The patient was treated in other institution and referred in our (institution -tertiary abdominal surgery center). We had only writing data about her treatment --without adjuvant therapy after first operation. 
  • Because of that we cannot answer to You.
  • The first sentence is inserted into last version of the submitted manuscript.

  1. How was the literature searched ? (I expected here features of a systematic review methodology so that someone could repeat their search.)  

  • A search of the following online databases and search engines were performed (PubMed/Medline, Web of Science and Embase) completed by an internet-based search. The keywords used were “endometrial, cancer, spleen, metastases”. All references of included articles were manually screened to find relevant articles.
  • This is also inserted into last version of the submitted manuscript.

  1. The MS was not checked by a native English speaker.

  • Manuscript was checked by Filipodia Science Management & Communication Experts publishing service (Email: [email protected]) with certificate paid 200 USD and certificate from 23. 10. 2021 (which we can send to You). The edit has achieved “Grade A: priority publishing; no language polishing required after editing”.

MINOR issues -we corrected

- Line 70, the keyword ‘Isolated’ is not in line with MeSH recommendations: : https://meshb.nlm.nih.gov/search;  

  • deleted

- Line 79, is: ‘ … in the literature[6,7,8-12].’; why not ‘ … in the literature[6-12].’; 

  • OK

- from Line 87 on, why is the ‘Case Presentation’ divided into subchapters and not one paragraph ?

  • Is it strictly forbidden? Technical editor could erase subheadings in the second.

- Line 102, the phrase ‘hydro electrolyte status’ is not English;

  • We are yet trying in medicine to speak Latin language, but it is corrected.

- Line 141, is: ‘ … Alcian Blue-Periodic acid-Schiff, … ‘; the literature rather refers to this staining as: periodic acid—Schiff (PAS)—Alcian Blue;  

- Lines 169-170: the number of CTX cycles is not given;

  • Number was given -6 cycles in original version

- Line 205, ‘sinchronous’ is misspelled.  

  • OK

FINALLY,  IF YOU THINK THAT OUR MANUSCRIPT ABOUT SUCCESSFUL MANAGEMENT OF THE VERY RARE CASE (15TH IN THE WORLD), IS NOT VALUABLE, YOU CAN REJECT IT FROM JOURNAL.

Sincerely

The authors of the manuscript “ISOLATED SPLEEN METASTASES OF ENDOMETRIAL CANCER- A case report”